# Thermal phenomena and size effects of Mg powder in combustion process

Ki-Hun Nam[1], Jung Kyu Park[2]*, Jun-Sik Lee[3]*

1 Department of Fire and Disaster Prevention Engineering, Changshin University, Changwon-si, Korea,
2 Department of Computer Engineering, Daejin University, Pocheon-si, Korea, 3 Department of Aviation Maintenance & Mechanical Engineering, Changshin University, Changwon-si, Korea

☯ These authors contributed equally to this work.
* jkpark@daejin.ac.kr (JKP); jslee@cs.ac.kr (JSL)

**Data Availability Statement:** All relevant data are within the manuscript and its Supporting information files. In our experiments, commercial Mg powder was used. The following link contains

## Abstract

Magnesium is a combustible metal that poses various safety risks, including fires and explosions. However, there are limited safety measures available to prevent and respond to potential fires and explosion incidents in the metal industry. In this study, the combustion process of Mg fires was closely examined using infrared thermal imaging, focusing on the effects of Mg powder size. For the experiment, Mg powder was burned by increasing the temperature to approximately 967.4 K using an ignition unit and controller equipped with a tungsten heater. Moreover, combustion velocity measurement experiments for Mg particle sizes of 75, 100, and 150 $\mu m$ were conducted using the combustion velocity measurement device presented in the NFPA 484 standard. On combustion of Mg, flames are observed; smoke is emitted as demonstrated by thermal and flow visualization experiments. The combustion velocity measurement experiment results demonstrated that the greater the slope value (combustion velocity) for the combustion length over time, the faster is the combustion velocity, with the 75 $\mu m$ particle size having the fastest combustion velocity. The results of this experiment can be utilized as references for Mg fire control design and to gain a better understanding of the scope of smoke and fire hazard investigation measures.

## Introduction

Magnesium and its alloys are widely used as key materials in the aircraft, automobiles, IT, and electronics field owing to their good specific strength and excellent formability, which are superior to those of aluminum and iron [1–4]. Moreover, the low melting point of Mg facilitates easy recycling, making it an eco-friendly metal; hence, its use continues to expand [5, 6]. However, Mg is a combustible metal that readily reacts with oxygen in the air, posing a significant risk of fire and explosion [7]. Despite the increasing utilization of Mg, research on preventive measures to address the increasing number of fires related to Mg is scarce. In particular, the ignition point of Mg varies depending on the particle size; that is, smaller particles are more prone to fires and explosions [4, 7–10]. A previous study that investigated the characteristics of flame propagation mechanisms based on the average diameter sizes (60, 170, 360, and 500 $\mu m$) through thermal imaging analysis found that the flame propagation speed increased

information on the powder used in the experiment. https://hanaamt.com/metal-powder/mg-metal-powder/.

**Funding:** This work was supported by the National Research Foundation of Korea (NRF) grant funded by the Korean Government (MSIT) (No. 2021R1F1A1055898) This work was supported by the National Research Foundation of Korea (NRF) grant funded by the Korean Government (MSIT) (No. 2022R1F1A1074289).

**Competing interests:** NO authors have competing interests.

as the average particle diameter decreased. The mechanism of flame propagation over a layer of Mg metal powder was analyzed in detail using infrared thermal imaging to measure the surface temperature of the powder layer [8], with a case study showing the combustion sequence in the order of (1) pre-combustion, (2) combustion (generating white smoke), (3) and post-combustion stages. Furthermore, research on the ignition temperature of Mg powder indicates that the reaction temperature decreases as the surface area increases [11]. A previous study investigated the oxidation and ignition behavior of Mg alloys containing rare-earth elements [12].

Although extensive research has been performed on the combustion characteristics and performance of Mg powder, studies focusing on post-ignition combustion and flow phenomena are still lacking. Therefore, this study aims to determine the combustion characteristics of Mg metal powder by analyzing its thermal properties using an infrared thermal imaging camera and thermocouples. The results of this experiment investigations can provide valuable insights into extending smoke and hazard fire investigation measures; they can also be used as a reference for magnesium fire control and response design.

## Materials and methods

### Properties of Mg powder

In this study, Mg of purity 99.5% or higher, as specified by the International Organization for Standardization 7165 (ISO 7165) standards for performance testing of fire extinguishers for metal fires was used [13]. Table 1 lists the physical properties of Mg metal, which is a light, silvery-white metal [13–15], as shown in Fig 1. Its specific gravity is 1.74, making it the lightest metal, and it has a relatively low melting point of 922.15 K and boiling point of 1373.15 K. The auto-ignition point varies depending on the particle size. It tends to be lower for smaller particles [13, 16]. The auto-ignition point of Mg powder is 745.93 K and its boiling point is approximately 1,373.15 K, as listed in Table 1. The Mg powder used in this study has an auto-ignition point of approximately 743.15 K because the size of Mg powder is smaller than the reference value in Table 1. The particle sizes of Mg used in this study were 150, 100, and 75 $\mu m$. ISO-7167 enacted that all particles should be able to pass a sieve with 387 $\mu m$ mesh size, and a minimum of 80% of the powder must be retained in a sieve with 150 $\mu m$ mesh size [13]. For this reason, the particle size was selected step by step based on the magnesium particle size of 150 $\mu m$ used in the ISO 7165 fire extinguisher performance test for metal fires [16].

### Experimental set-up and conditions

This study investigates the phenomenon of Mg metal fire based on thermal and flow visualization because the smoke cannot be ocularly observed during Mg metal burning. The experimental equipment comprises a diode-pumped solid-state (DPSS) laser, a high speed camera, and an IR camera (FLIRA655sc). After the DPSS laser forms a 2D plane, a CCD camera is used to capture the image flowing on the plane. Additionally, the combustion velocity of Mg was measured using a combustion velocity measurement device based on the particle size. The

**Table 1. Material properties of magnesium [15].**

| Autoignition point (K) | | | Specific gravity |
|---|---|---|---|
| Powder | Ribbons and shavings | Chunk | 1.74 |
| 745.93 | 783.15 | 923.15 | |

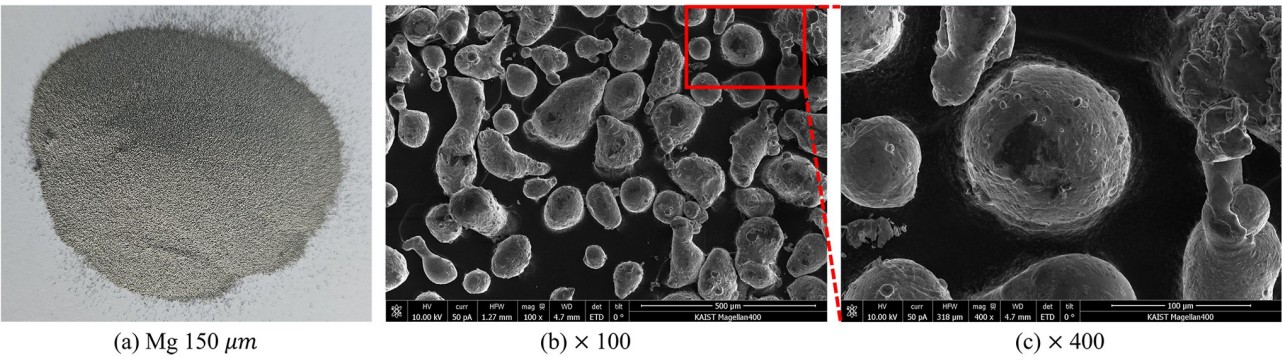

(a) Mg 150 $\mu m$ (b) × 100 (c) × 400

**Fig 1. Magnesium powder (150 $\mu m$(a)) used in the experiment (FE-SEM images × 100 (b) and × 400 (c)).**

characteristics of the combustion product particles were analyzed using SEM (model: S-4800 made by Hitachi).

Figs 2 and 3 show the schematic of the flow and thermal visualization setup for the smoke of Mg metal fire. The setup consists of a DPSS laser, a smoke generator, an IR camera, and a video camera. A digital camera was used to capture the flow images that were illuminated by a laser sheet. Simultaneously, the IR camera, which could operate up to a temperature of 2273.15 K, captured thermal images. A data logger (model GL840) and four K-type thermocouples (0.32 mm, 3.15–1, 545.15 K) were utilized to calibrate and measure the combustion temperature of Mg powder.

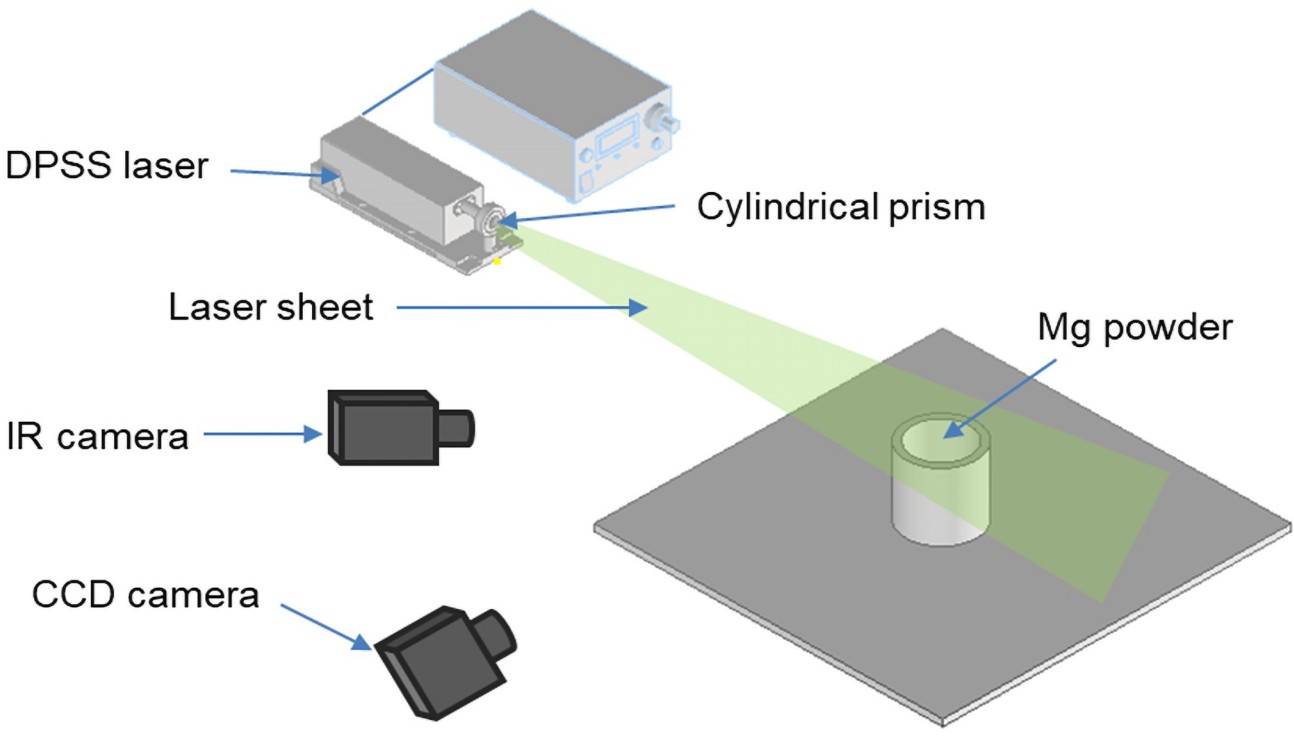

**Fig 2. Schematic of flow visualization setup.**

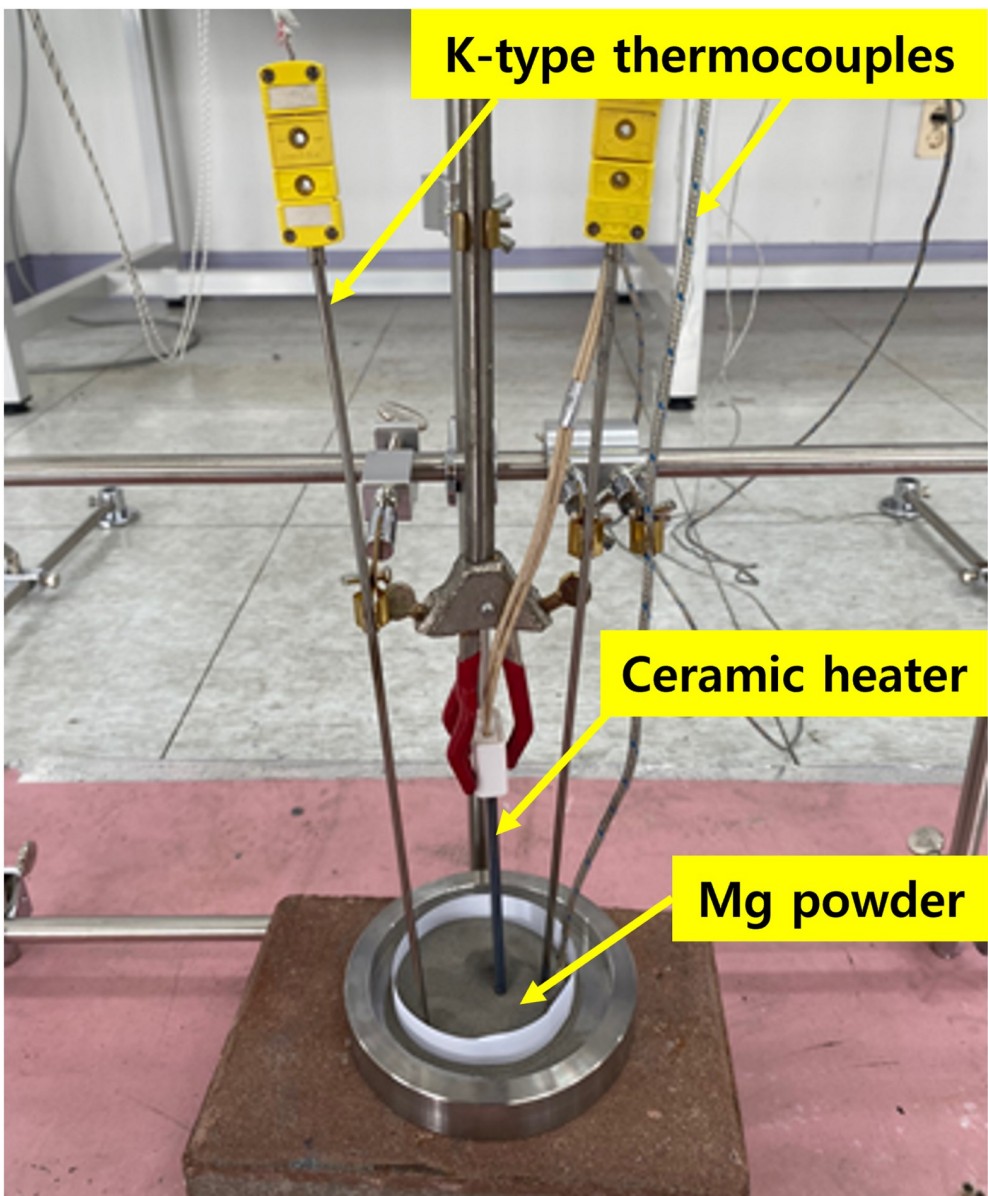

**Fig 3. Schematic of thermal visualization setup.**

In this experiment, 20 g of Mg powder with an average particle size of 75 $\mu$m and a purity of over 99% was utilized, as shown in Fig 2. Before the experiment, the Mg powder was adequately dried by storing it in a desiccator for 24 h at a temperature of 296.15 K ± 278.15 K. The experimental setup was prepared, and the Mg powder was ignited using an ignition device (ceramic heater) for approximately 5 min. Experimental data were simultaneously collected through the IR camera, thermocouples, and flow visualization apparatus. Despite only a small amount of Mg used for the combustion experiments, safety measures were taken, as shown in Fig 3, by installing a 10-mm-thick gypsum board on the floor, on top of which the experimental bed and a circular container were placed. During the combustion process, Mg continued to burn after oxygen was depleted. Subsequently, Mg reacted with nitrogen from the air to form

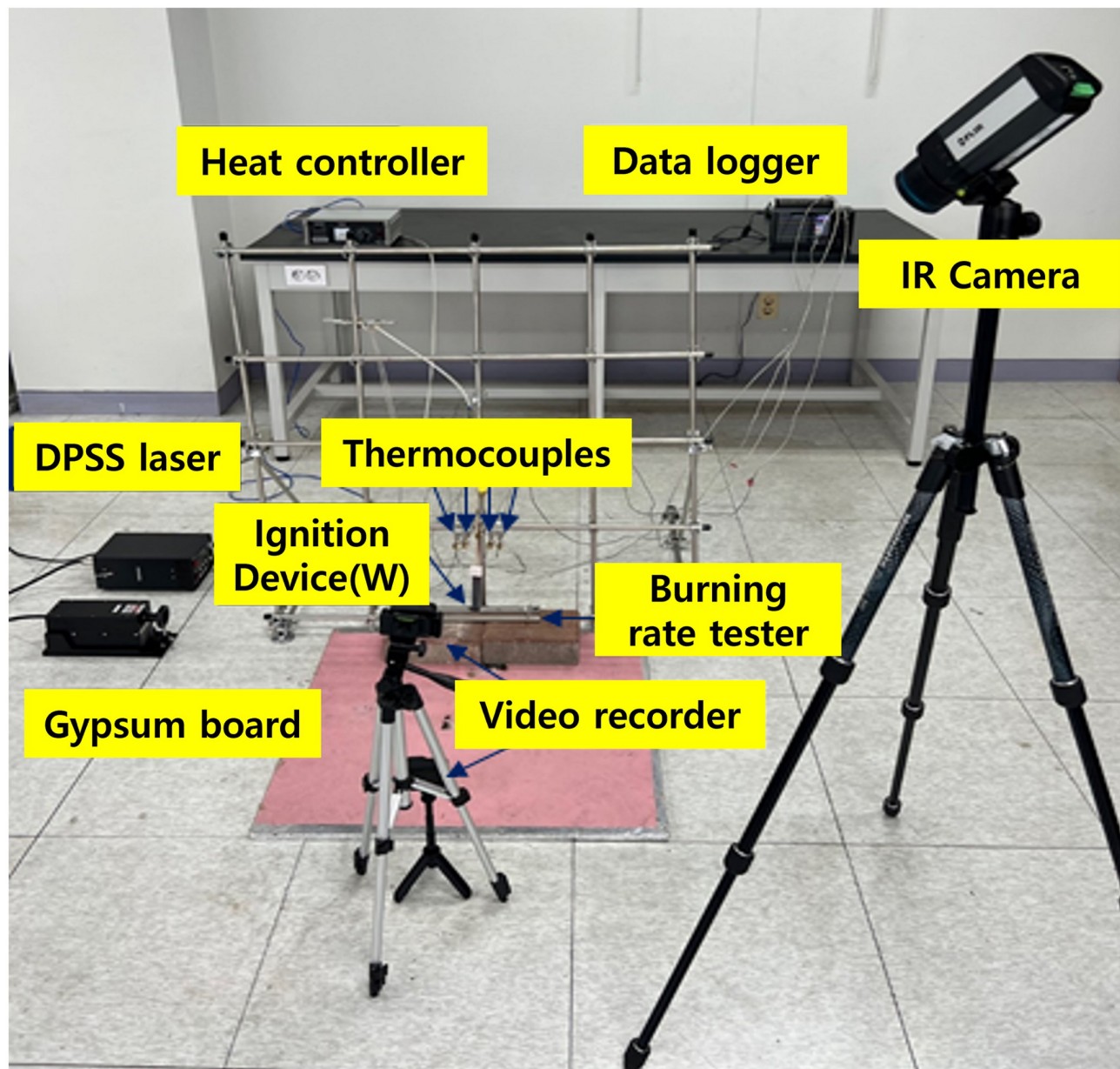

**Fig 4. Set up of the burning rate experiment equipment.**

$Mg_3N_2$. Following the methodology outlined in [17], the experiment on flow visualization was conducted to visualize the height of flame propagation caused by natural convection within the chamber.

To understand the combustion velocity of Mg powder, a combustion velocity measurement device was made based on the standards specified in the NFPA 484 standard using an ignition device, as shown in Fig 4. Combustion experiments according to Mg powder particle size were conducted nine times, with each of the three sizes of Mg powder tested three times. For the combustion experiment, 10 g of Mg particles of sizes 150, 100, and 75 $\mu m$ were placed in the combustion velocity measurement device and heated using the ignition device for 5 minutes.

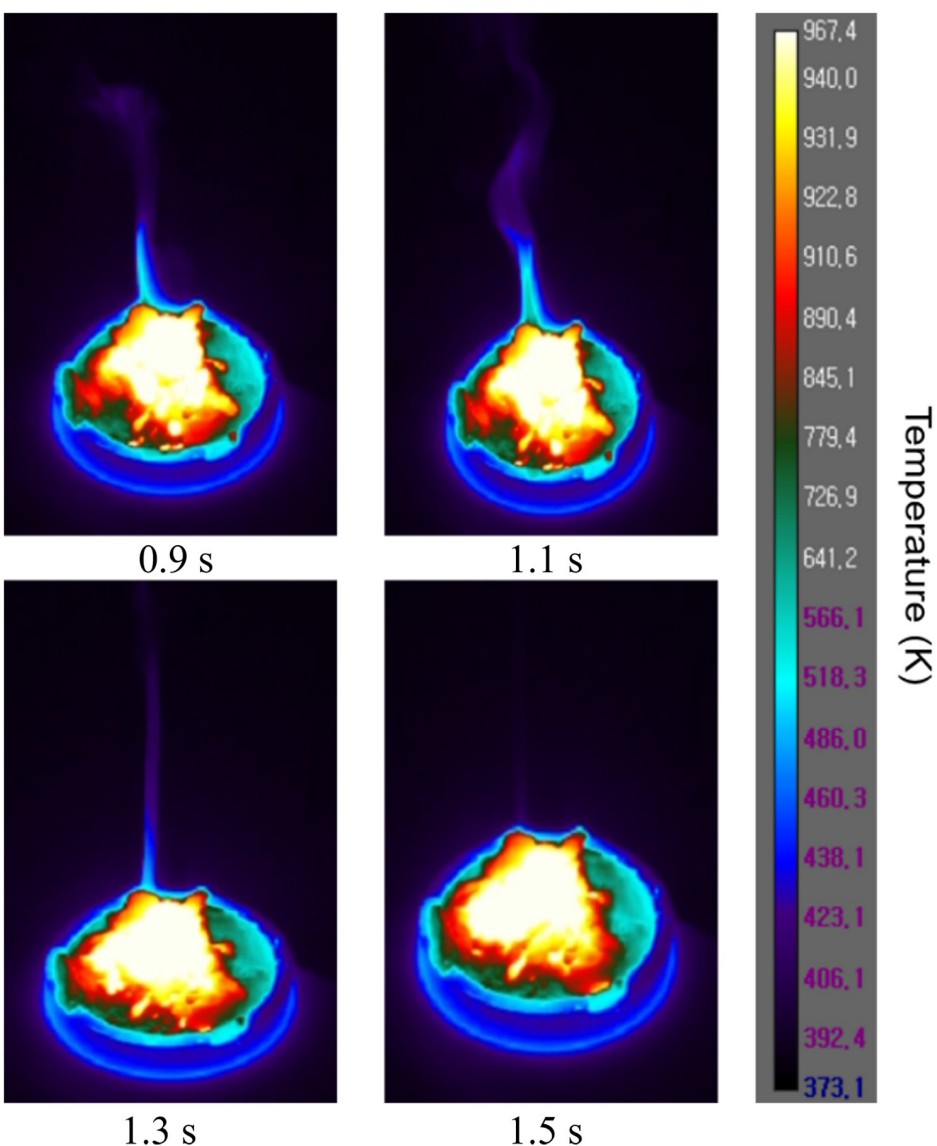

**Fig 5. Infrared thermal image of Mg combustion.**

## Results and discussion

Fig 5 show infrared images of the temperature reached after ignition and thermal images depicting the Mg combustion process over time. Owing to the high-temperature conditions under which the experiment is conducted, the temperature range spans from 373.1K to 967.4K to observe heat flow phenomena during combustion. The images show the flame spreading due to the combustion heat. There was a rapid burning accompanied by a white flame and white smoke. Upon analyzing the temperature distribution through the infrared thermal images, we observed that the peripheral areas where combustion is accompanied by flame show a temperature distribution above 967.4 K. Fig 6 shows the temperature changes from the time of ignition to the time of reaching maximum temperature, measured at the center of the Mg powder surface by the thermocouples. The maximum temperature reached approximately

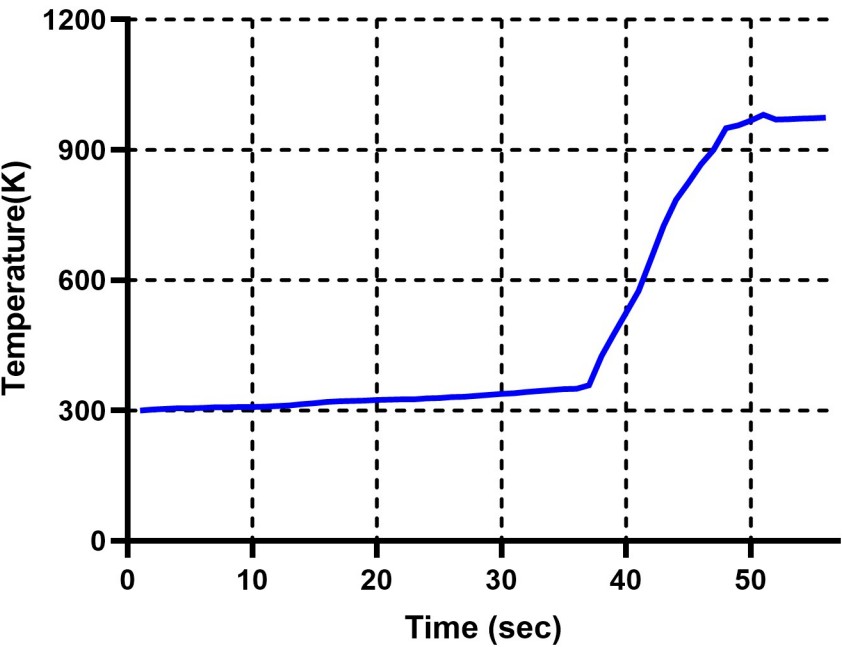

**Fig 6. Temperature of the thermocouple on the surface of Mg.**

993.15 K, 48 s after ignition, followed by a trend of small decreases and increases. There are slight discrepancies between the temperatures shown in the thermal images and the temperatures measured by the thermocouples. This is because the temperature differs with the measurement location (surface temperature). Furthermore, the heat generated during combustion does not transfer internally beyond the boiling point of Mg, affecting the ongoing combustion process.

### Thermal and fluid flow characteristics

Fig 7 presents a flow visualization of the combustion approximately 1.5 s after ignition. We can observe the smoke flow pattern of the burning Mg powder. During the combustion process, intense light and heat are generated in the middle portion, and the flame temperature of burning Mg reaches up to approximately 3,273.15 K [14], rendering it unrecordable by optical cameras. As the combustion of Mg progresses, the flame temperature rises above the boiling point (1,380.15 K). At this point, the vaporized Mg on the surface continues to combust as it combines with oxygen, producing MgO. The white smoke generated during combustion (shown as green smoke in Fig 7) is solid particles of oxidized Mg produced as Mg vaporizes and combusts; these particles solidify due to the high melting point of Mg.

### Combustion velocity

Mg demonstrates a higher activation energy and lower reactivity when the metal particles are larger. When the particle size is small, the temperature increase is faster and the preheating time is shorter than in larger particles, causing the reaction to occur more quickly [2, 5, 8, 9].

Fig 8 shows the combustion length over time by the Mg particle size. The combustion velocity measurement results indicate that on average 90, 60, and 45 s was required for 150, 100, and 75 $\mu$m of Mg to burn 30 mm, respectively. This indicates that the smaller the Mg particle

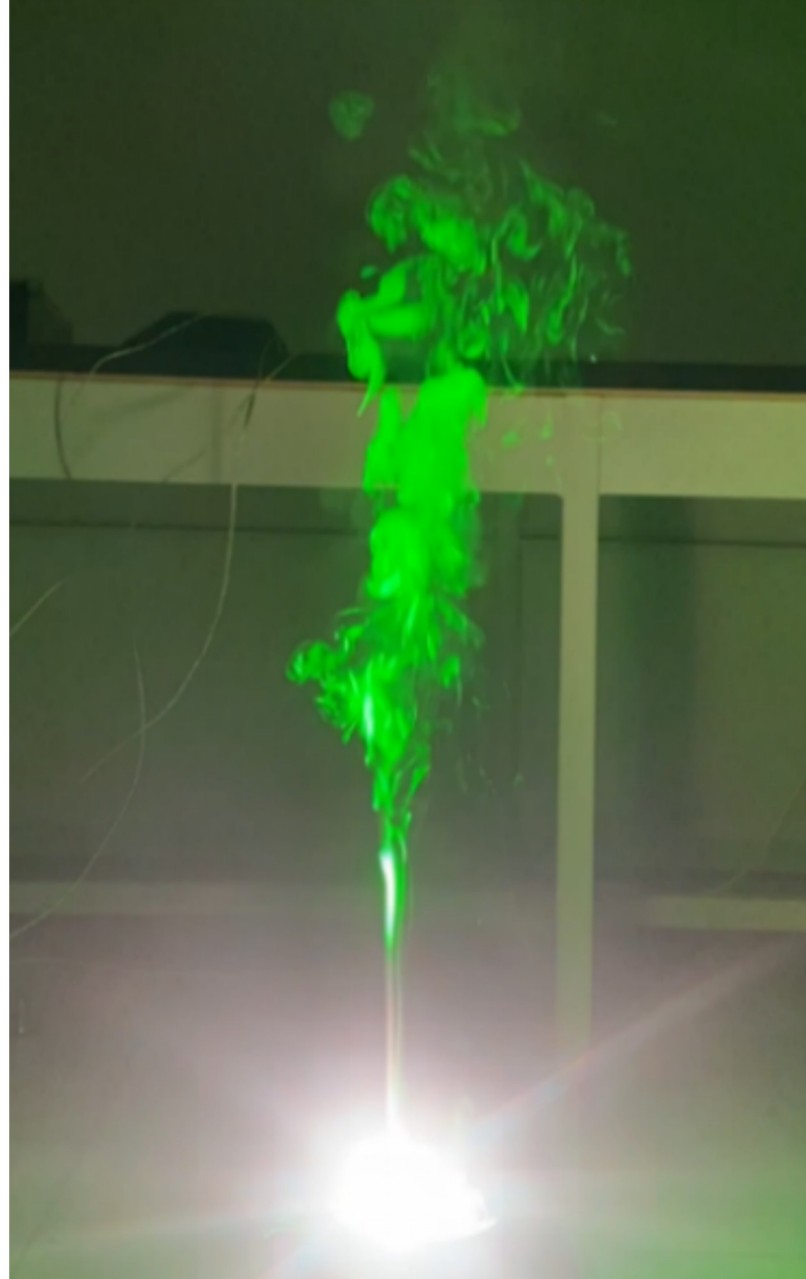

**Fig 7. Photo of experimental flow visualization 1.5 s after ignition.**

size, the faster is the combustion velocity and the quicker the combustion spread. Additionally, after combustion, the average length of the combustion residue was 32, 35, and 45.7 mm for Mg 150, 100, and 75 $\mu m$, respectively. The slope of the combustion length over time was 9.2, 8.0, and 9.5 for 150, 100, and 75 $\mu m$, respectively. A higher slope value indicates a faster combustion velocity, with 75 $\mu m$ being the fastest. This suggests that the smaller the Mg particle size, the faster is the heat diffusion among particles during combustion, and that fires involving smaller Mg particles may be more hazardous than those involving larger Mg particles.

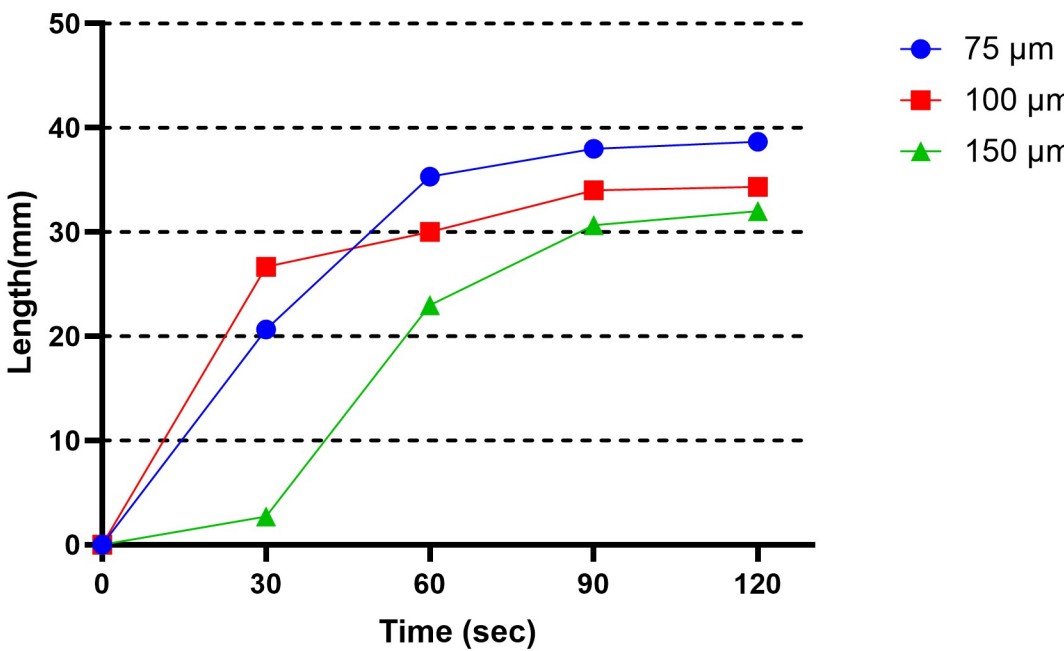

**Fig 8. Combustion velocity by particle size.**

The change in the particle shape and composition after Mg combustion was analyzed through SEM photography as shown in Fig 9. Mg has an irregular shape, mostly forming plates, and smaller sizes tend to be more spherical. Additionally, there was no agglomeration among particles, which is common in fine particles before combustion. However, compared to the circled and irregular shapes in the pre-combustion of magnesium, the samples appear in varying forms.

Magnesium undergoes melting, evaporation, and combustion processes due to heat. During these processes, magnesium goes through a melting process and combines with oxygen in the air to produce MgO, and reacts with nitrogen to produce $Mg_3N_2$. In addition, a reaction occurs to generate MgOH by combining with some atmospheric moisture. In this melting and chemical reaction process, the reaction rate and amount of reaction vary depending on the particle size, leading to different types of combustion products and particle shapes [18, 19].

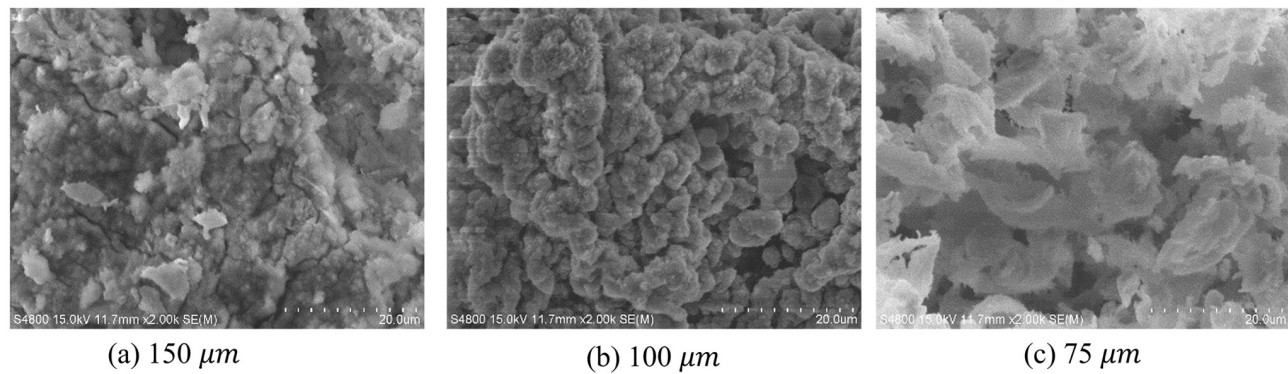

(a) 150 $\mu m$      (b) 100 $\mu m$      (c) 75 $\mu m$

**Fig 9. SEM images of Mg combustion products.**

Due to this phenomenon. Unlike pure magnesium, which showed a non-uniform but round shape as shown in Fig 1(b) and 1(c), after combustion, it is believed that various inconsistent changes in shape appeared depending on the size of the particles. There is no significant difference in the transformation of magnesium particles by the high temperature heat of combustion.

## Conclusion

This study experimentally investigated the combustion characteristics and fire development of Mg through thermal and flow visualization. The thermal and flow visualization results demonstrate smoke-developing patterns during Mg metal fire. The combustion reaction was not visible initially but was observed at the smoldering and fire stages. It transformed to flame in the fire stage and eventually smoldered. This experiment showed that smaller Mg particles have faster heat diffusion during combustion, indicating a higher risk if a fire occurs with smaller Mg particles. Therefore, developing management strategies and fire suppression technologies that consider the impact of Mg particle size on fire spread is necessary. After measuring the combustion rate according to the particle size, the time for the combustion length to reach 30 mm was 90 seconds for 150 $\mu m$, 60 seconds for 100 $\mu m$, and 45 seconds for 75 $\mu m$. The average combustion length for each size was 32 mm for 150 $\mu m$, 35 mm for 100 $\mu m$, and 45.7 mm for 75 $\mu m$. This indicates that the smaller the particle size, the faster the combustion and diffusion rate. Additionally, in Figs 5 and 6, a rapid increase in temperature occurred at approximately 37 seconds and reached the maximum temperature at approximately 48 seconds. As shown in Fig 8, this means that magnesium changes to a vapor state through a smoldering process after ignition, generating a flame and rapid combustion. The results of this experiment can also be utilized to better formulate and expand smoke prevention measures for Mg-related fire hazards. In addition, the results can serve as references for the design of metal fire control. Notably, research on Mg fires is still in its early phases. In the future, we intend to investigate the mechanism of combustible Mg metal fire. Furthermore, we plan to further explore the combustion characteristics and toxicity with increasing amounts of Mg to propose solutions for extinguishing agents and responsive technologies.

## Supporting information

**S1 File. MSDS of Mg powder.** Material Safety Data Sheet of Magnesium Powder.
(PDF)

**S2 File. Graph data.** Raw graph data of Figs 5 and 7.
(XLSX)

## Author Contributions

**Supervision:** Jung Kyu Park, Jun-Sik Lee.

**Validation:** Jun-Sik Lee.

**Writing – original draft:** Ki-Hun Nam.

**Writing – review & editing:** Jung Kyu Park, Jun-Sik Lee.

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
