## [Decision Letter · Decision Letter 0]

23 Jun 2024

PONE-D-24-21781Thermal Phenomena and Size Effects of Mg PowderPLOS ONE

Dear Dr. Jung Kyu Park,

Thank you for submitting your manuscript to PLOS ONE. After careful consideration, we feel that it has merit but does not fully meet PLOS ONE’s publication criteria as it currently stands. Therefore, we invite you to submit a revised version of the manuscript that addresses the points raised during the review process.

**The author need to work on the following comments as ****1. Provide the characterization study with details explanation.****2. Improve the quality of the figures.****3. Enlist the application of the present work.**4. Add the highend characterization study images.5. Elaborate the results and discussion in details.6. Rewrite the conclusion section.7. Follow the journal template for reference section.==============================

We look forward to receiving your revised manuscript.

Kind regards,

Sameer Sheshrao Gajghate, PhD

Academic Editor

PLOS ONE

Journal Requirements:

"This work was supported by the National Research Foundation of Korea (NRF) grant funded by the Korean Government (MSIT) (No. 2021R1F1A1055898 & No. 2022R1F1A1074289)"

"NO authors have competing interests"

4. We note that your Data Availability Statement is currently as follows: All relevant data are within the manuscript and its Supporting Information files

Reviewers' comments:

Reviewer's Responses to Questions

**Comments to the Author**

1. Is the manuscript technically sound, and do the data support the conclusions?

Reviewer #1: Partly

Reviewer #2: Yes

2. Has the statistical analysis been performed appropriately and rigorously? 

Reviewer #1: Yes

Reviewer #2: N/A

3. Have the authors made all data underlying the findings in their manuscript fully available?

Reviewer #1: Yes

Reviewer #2: Yes

4. Is the manuscript presented in an intelligible fashion and written in standard English?

Reviewer #1: No

Reviewer #2: Yes

5. Review Comments to the Author

Reviewer #1: 1. What is the novelty in the manuscript?

2. In title, this work focused on the results of thermal and size on Mg powder in combustion process, so, it is suggested to revise as: Thermal Phenomena and Size Effects of Mg Powder in Combustion Process.

3. In application industries, what is the range of size in Mg powder? In this manuscript, why did select the size for 75, 100, and 150 μm?

4. Figures are not clear, please label their numbers. Moreover, “Fig 4 and Fig 5 show infrared images of the highest temperature reached after ignition and thermal images depicting the Mg combustion process over time.” The images are in-situ combustion process? It should be selected different temperatures at the different stages in the combustion process, highlighting their stage characteristics and key parameters. Furthermore, the image should be described before the Mg powder (raw material) as the control group.

5. In section Results and discussion, the contents and results should be deeply described and thoroughly interpretated.

6. The references should be updated and supplemented.

7. The conclusion should be rewritten, for supplementing some quantization results.

Reviewer #2: The subject of the manuscript focused on thermal phenomena and size effects of Mg powder is in good relevance with the scope of PLOS ONE.

The introduction properly presents the issues related to post-ignition combustion and flow phenomena. Materials used as well as the experimental set-up and conditions are described. However, this description must be reorganized because some of the data regarding the research methodology was discussed in the Introduction section, and detailed data was not provided in the Materials and Methods section. This mainly concerns the characterization of the combustion product particles using SEM. A description of the conditions for testing the structure of Mg particles should be provided.

Results and discussion part concerning the results obtained for the combustion characteristics and fire development of Mg through thermal and flow visualization is sufficiently detailed. However, the caption for Fig. 9 is missing. Moreover, the description of the SEM results should also include a reference to Fig. 1 illustrating the lack of agglomeration of Mg particles, which is less visible in Fig. 9.

6. PLOS authors have the option to publish the peer review history of their article (what does this mean?). If published, this will include your full peer review and any attached files.

Reviewer #1: No

Reviewer #2: No

---

## [Author Response · Author response to Decision Letter 0]

21 Aug 2024

Reviewer #1: 

Comment 1: What is the novelty in the manuscript?

-> Response 1: Thank you for your comments. We regret that the analysis does not have innovative analysis. We analyzed the risk depending on the particle size and the process of magnesium combustion. Through the risk analysis above, we focused on obtaining the scientific data necessary to prevent and respond to magnesium fires. The analysis was additionally written in the paper.

Comment 2: In title, this work focused on the results of thermal and size on Mg powder in combustion process, so, it is suggested to revise as: Thermal Phenomena and Size Effects of Mg Powder in Combustion Process.

-> Response 2: Thank you for your comments. We agree with this comment. We have changed the research title. 

From: Thermal and flow characteristic of magnesium powder combustion

To : Thermal Phenomena and Size Effects of Mg Powder in Combustion Process.

Comment 3 In application industries, what is the range of size in Mg powder? In this manuscript, why did select the size for 75, 100, and 150 μm?

-> Response 3: We appreciate you pointing this out here. I think this point is an important part in this manuscript. Following your comments, we explain why we selected the sizes. 

Comment #4 Figures are not clear, please label their numbers. Moreover, “Fig 4 and Fig 5 show infrared images of the highest temperature reached after ignition and thermal images depicting the Mg combustion process over time.” The images are in-situ combustion process? It should be selected different temperatures at the different stages in the combustion process, highlighting their stage characteristics and key parameters. Furthermore, the image should be described before the Mg powder (raw material) as the control group.

-> Response 4: Thank you for your comments. We deleted Figure 4 because it was not clear.

And we comprehensively analyzed Fig 5 and Fig 6 and wrote additional content.

Fig 4 : deleted and Fig 5 and Fig 6 is renamed to Fig 4 and Fig 5

Comment 5: In section Results and discussion, the contents and results should be deeply described and thoroughly interpretated.

-> Response 5: Thank you for your comments. We have rewritten the contents in the Results and Discussion section. We have written additional content in the Combustion Velocity subsection.

Comment 6: The references should be updated and supplemented.

-> Response 6: We add and update some references. 

Manju M. Explosion characteristics of micron- and nano-size magnesium powders. J. Loss Prevent. Proc. 2014, 27, 55–64.

K.H. Nam, J.S. Lee, & H.J. Part. Understanding Combustion Mechanism of Magnesium for Better Safety Measures: An Experimental Study. Journal of Safety, 2022, 8, 11 

Li, G.; Yuan, C.; Zhang, P.; Chen, B. Experiment-based fire and explosion risk analysis for powdered magnesium production methods. J. Loss Prevent. Proc. 2008, 21, 461–465.

Yuan, C.; Yu, L.; Li, C.; Li, G.; Zhong, S. Thermal analysis of magnesium reactions with nitrogen/oxygen gas mixtures. J. Hazard. Mater. 2013, 260, 707–714

Comment 7: The conclusion should be rewritten, for supplementing some quantization results.

-> Response 6: We have written a new paragraph that can supplement the content in the conclusion.

Fig 1, 3, 8 Replaced with high-resolution image (eps format)

Fig 5. Create a new graph (eps format)

Reviewer #2: 

Comment 1: However, this description must be reorganized because some of the data regarding the research methodology was discussed in the Introduction section, and detailed data was not provided in the Materials and Methods section.

-> Response 1: Additional information on the experimental substance magnesium and the experimental method were explained.

Comment 2: This mainly concerns the characterization of the combustion product particles using SEM. A description of the conditions for testing the structure of Mg particles should be provided.

-> Response #2: We added a detailed description of the experimental equipment including SEM used in the experiment.

Comment 3: Results and discussion part concerning the results obtained for the combustion characteristics and fire development of Mg through thermal and flow visualization is sufficiently detailed. 

-> Response 3: We have rewritten the contents in the Results and Discussion section. We have written additional content in the Combustion Velocity subsection. We have written a new paragraph that can supplement the content in the conclusion.

Comment 4: However, the caption for Fig. 9 is missing. Moreover, the description of the SEM results should also include a reference to Fig. 1 illustrating the lack of agglomeration of Mg particles, which is less visible in Fig. 9.

-> Response #4: Thank you for your comments. We comprehensively analyzed Fig 1 and Fig 8 and wrote additional content.

Fig 9 is renamed Fig 8.

Fig 1, 3, 8 Replaced with high-resolution image (eps format)

Fig 5. Create a new graph (eps format)

Thank you for your consideration. We look forward to hearing from you.

---

## [Decision Letter · Decision Letter 1]

27 Aug 2024

Thermal Phenomena and Size Effects of Mg Powder in Combustion Process

PONE-D-24-21781R1

Dear Dr. Jung Kyu Park,

We’re pleased to inform you that your manuscript has been judged scientifically suitable for publication and will be formally accepted for publication once it meets all outstanding technical requirements.

Kind regards,

Sameer Sheshrao Gajghate, PhD

Academic Editor

PLOS ONE

Additional Editor Comments (optional):

Reviewers' comments:

Reviewer's Responses to Questions

**Comments to the Author**

1. If the authors have adequately addressed your comments raised in a previous round of review and you feel that this manuscript is now acceptable for publication, you may indicate that here to bypass the “Comments to the Author” section, enter your conflict of interest statement in the “Confidential to Editor” section, and submit your "Accept" recommendation.

Reviewer #1: All comments have been addressed

Reviewer #2: All comments have been addressed

2. Is the manuscript technically sound, and do the data support the conclusions?

Reviewer #1: Yes

Reviewer #2: Yes

3. Has the statistical analysis been performed appropriately and rigorously? 

Reviewer #1: Yes

Reviewer #2: N/A

4. Have the authors made all data underlying the findings in their manuscript fully available?

Reviewer #1: Yes

Reviewer #2: Yes

5. Is the manuscript presented in an intelligible fashion and written in standard English?

Reviewer #1: Yes

Reviewer #2: Yes

6. Review Comments to the Author

Reviewer #1: My comments have been responded in detailed, this manuscript can be recommended to publish in PLOS ONE.

Reviewer #2: The subject of the manuscript focused on thermal phenomena and size effects of Mg powder is in good relevance with the scope of PLOS ONE.

The introduction properly presents the issues related to post-ignition combustion and flow phenomena. Materials used as well as the experimental set-up and conditions are described. The characterization of the combustion product particles using SEM was completed. A description of the conditions for testing the structure of Mg particles was provided.

Results and discussion part concerning the results obtained for the combustion characteristics and fire development of Mg through thermal and flow visualization is sufficiently detailed. In the Results and Discussion section, the figures were reorganized and appropriate descriptions and comments were added.

The quality of the manuscript is good.

7. PLOS authors have the option to publish the peer review history of their article (what does this mean?). If published, this will include your full peer review and any attached files.

Reviewer #1: No

Reviewer #2: **Yes: **Maria Zielecka

---

## [Editor Report · Acceptance letter]

4 Sep 2024

PONE-D-24-21781R1 

PLOS ONE

Dear Dr. Park, 

I'm pleased to inform you that your manuscript has been deemed suitable for publication in PLOS ONE. Congratulations! Your manuscript is now being handed over to our production team.

Kind regards, 

on behalf of

Dr. Sameer Sheshrao Gajghate 

Academic Editor

PLOS ONE